# Clinical Efficacy and Safety of Two Cycles of Intra-Articular Injection of Porcine Atelocollagen Versus Hyaluronic Acid in Knee Osteoarthritis

**DOI:** 10.3390/bioengineering12070710

**Published:** 2025-06-29

**Authors:** Yong In, Keun Young Choi, Man Soo Kim

**Affiliations:** Department of Orthopaedic Surgery, Seoul St. Mary′s Hospital, College of Medicine, The Catholic University of Korea, 222, Banpo-daero, Seocho-gu, Seoul 06591, Republic of Korea; iy1000@catholic.ac.kr (Y.I.); heaxagon@hanmail.net (K.Y.C.)

**Keywords:** knee osteoarthritis, intra-articular injection, porcine atelocollagen, hyaluronic acid, non-inferiority

## Abstract

(1) **Background:** Knee osteoarthritis (KOA) induces pain, stiffness, and impaired mobility, particularly in aging populations. Despite providing symptom relief, the long-term efficacy of intra-articular hyaluronic acid (HA) injections remains unclear. With its longer intra-articular residence time and potential chondroprotective effects, porcine-derived atelocollagen is an alternative to HA. We aimed to compare the safety and efficacy of collagen versus HA injections in symptomatic KOA. (2) **Methods:** This retrospective observational study included 40 patients with KOA who received either two cycles of collagen or HA injections at 6-month intervals. Clinical outcomes were assessed using the visual analog scale (VAS) and the Western Ontario and McMaster Universities Osteoarthritis Index (WOMAC) at baseline and 6 months after the first and second injections (Cycle 1 and Cycle 2, respectively). Patient satisfaction and adverse events were recorded. Non-inferiority analysis was conducted for VAS and WOMAC score changes. (3) **Results:** Significant intragroup improvements in VAS and WOMAC scores were noted after each injection cycle (*p* < 0.05), albeit without significant between-group differences, non-inferiority of collagen to HA based on predefined margins, and comparable patient-reported satisfaction (>85% reported improvement after each cycle), with similar incidence of mild adverse events (collagen: 20%, HA: 25%, *p* = 0.705). (4) **Conclusions:** Intra-articular collagen injections were clinically non-inferior to HA in reducing pain and improving function in patients with KOA across two treatment cycles. Given its favorable safety profile and potential structural benefits, collagen may serve as a viable alternative injectable therapy for the non-surgical management of KOA.

## 1. Introduction

Osteoarthritis (OA) is a degenerative joint disease characterized by progressive degradation of articular cartilage, subchondral bone remodeling, and synovial inflammation [1,2,3,4,5]. The knee is particularly vulnerable to OA owing to its load-bearing function. Knee osteoarthritis (KOA) is a significant cause of pain, functional disability, and reduced quality of life in aging populations [6,7]. KOA pathophysiology involves complex biochemical and mechanical interactions that induce cartilage loss, osteophyte formation, and joint-space narrowing, which causes pain, stiffness, and impaired mobility [8].

Treatment for KOA comprises conservative options, such as lifestyle modifications and pharmacotherapy, and invasive surgical interventions, including osteotomy and arthroplasty [9]. Intra-articular injections are a minimally invasive alternative treatment for reducing pain, improving joint lubrication, and postponing surgery [9]. Hyaluronic acid (HA) injections are widely used owing to their viscoelastic properties, which enhance synovial fluid viscosity and provide a cushioning effect [10]. Repeated intra-articular HA injections are effective and safe for KOA management, and subsequent treatment cycles consistently maintain and can further enhance pain relief without increasing adverse events [11].

Recently, intra-articular collagen injections have emerged as a promising alternative [12,13,14]. Collagen, a key structural component of articular cartilage, has a longer intra-articular half-life than HA and may exert additional therapeutic effects beyond lubrication, including chondroprotection and structural reinforcement of the cartilage surface [15,16,17,18]. Specifically, type I atelocollagen derived from porcine sources potentially augments the lamina splendens, the superficial zone of cartilage, and thereby enhances joint protection and potentially modulates cartilage metabolism [16,19]. Studies on collagen injections for KOA have either been single-arm or short-term investigations that lacked control groups and long-term outcome measures. The clinical relevance of repeated administration—considered essential for chronic conditions, such as OA—remains underexplored [12,13,14]. In addition, clinical evidence of direct comparisons of the efficacy and safety of collagen and HA injections, across multiple treatment cycles, is limited [14]. Although a previous trial has compared collagen and HA [20], this study often involved only a single injection cycle or had limited follow-up [20]. Our study is, to the best of our knowledge, the first to perform a comparative evaluation of two complete injection cycles of intra-articular collagen versus HA in a real-world clinical setting. We also conducted a formal non-inferiority analysis, which has not been rigorously applied in prior comparative studies.

In this study, we aimed to compare the clinical efficacy and safety of collagen and HA after one and two injection cycles of these intra-articular therapies in a clinical context and explore whether collagen injection constitutes a viable alternative or adjunct to HA in routine orthopedic practice. Our results provide a rationale for informed decision-making for the non-surgical management of KOA.

## 2. Materials and Methods

### 2.1. Study Design and Ethical Approval

This retrospective comparative cohort study adhered to the ethical principles of the Declaration of Helsinki and was approved by the relevant Institutional Review Board. All participants provided written informed consent prior to enrollment.

### 2.2. Participants and Eligibility Criteria

We enrolled patients (age ≥ 40 years) who received two cycles of intra-articular injections of either HA or collagen at 6-month intervals, between 2023 and 2024, and were ambulatory after a diagnosis of KOA made at least 3 months prior to screening. Participants had experienced inadequate pain relief from prior treatment with analgesics or non-steroidal anti-inflammatory drugs. We screened electronic medical charts of adults and excluded patients with secondary OA or other inflammatory joint diseases (e.g., rheumatoid arthritis, psoriatic arthritis, or ankylosing spondylitis); significant joint deformity, instability, or effusion in the affected knee; a history of knee replacement surgery (total or partial) in the index joint; prior intra-articular injections (e.g., corticosteroids or hyaluronic acid) within 3 months before screening; a history of major surgery, arthroscopic intervention in either knee within the past 6 months, uncontrolled comorbidities (including cardiovascular disease or diabetes), active infections, or a history of malignancy within the past 5 years; or current use of systemic corticosteroids, anticoagulants, or immunosuppressive medications. To ensure comparability between the two groups, we applied the same inclusion and exclusion criteria, and selected patients consecutively over the same time period.

Among 86 patients identified in the HA group, we excluded 4 patients with secondary OA, 5 with prior knee surgery, 4 with prior intra-articular injections within 3 months, 4 with systemic corticosteroid or anticoagulants, 12 with uncontrolled comorbidities, 8 with a history of malignancy, and 29 who declined to participate, resulting in a final sample of 20 patients. Among 32 patients identified in the collagen group, we excluded 1 patient with prior knee surgery, 3 with uncontrolled comorbidities, 2 with a history of malignancy, and 6 who declined participation, leaving 20 patients in the final analysis. Both collagen and HA injections were administered to patients with Kellgren–Lawrence (KL) grade 2–3 KOA, which represents mild to moderate disease severity and is considered appropriate for intra-articular injectable therapies. The selection of injection type (collagen versus HA) was made through shared decision-making between the patient and the treating orthopedic surgeon. This process considered several clinical factors, including the patient’s prior response to HA, treatment expectations, financial considerations, and willingness to try novel biologic agents such as atelocollagen.

### 2.3. Interventions and Injection Protocol

Collagen injections consisted of either CartiZol Active (6% atelocollagen, 1 mL = 60 mg) or CartiZol Ultra (6% atelocollagen, 3 mL = 180 mg), both manufactured by Cellontech (Seoul, Republic of Korea). CartiZol Ultra was used once per 6-month cycle, whereas CartiZol Active could be administered up to three times within the same period at the surgeon’s discretion. HA injections employed Synovian, a 3 mL pre-filled syringe of 1,4-butanediol diglycidyl ether-cross-linked HA (LG Life Sciences, Seoul, Republic of Korea) delivered once per cycle. All injections were administered intra-articularly into the knee joint using a standard anterolateral approach, without ultrasound guidance, by an experienced orthopedic surgeon under aseptic conditions using a 22-gauge needle, with patients in a supine position. Patients were advised 24 h restriction of high-impact activity and ice application, in case of postprocedural discomfort; no prophylactic antibiotics were used.

### 2.4. Follow-Up Schedule and Outcome Measures

Data on demographic variables, such as age, sex, body mass index, and radiographic OA grade (based on the KL grade classification), were collected at baseline [21]. Comorbidities, such as hypertension, diabetes mellitus, thyroid disease, and smoking status, as well as side of injection (right or left knee) and product details, were recorded. Information on occupational activity was additionally collected as a proxy for physical functional status, and prior and ongoing use of analgesics, non-steroidal anti-inflammatory drugs (NSAIDs), and symptomatic slow-acting drugs for osteoarthritis (SYSADOAs); furthermore, a history of prior intra-articular injections was documented. Each knee was evaluated at baseline (pre-injection) and 6 months after the first and second injections (cycles 1 and 2), wherein a trained orthopedic surgeon documented pain scores using a 10 cm visual analog scale (VAS), subdivided into pain at rest, pain during walking, and nocturnal pain [22]. Functional status was ascertained with the validated Korean version of the Western Ontario and McMaster Universities Osteoarthritis Index (WOMAC), with subscores for pain, stiffness, and physical function (total score 0–96, with higher scores indicating worse status) [23]. Patient satisfaction was quantified at each follow-up visit using a five-point Likert scale as follows: “marked worsening,” “slight worsening,” “no change,” “slightly improved,” or “very improved.” For statistical analysis, responses were categorized into three levels: no change/worsened, slightly improved, and very improved. Adverse events were monitored through chart review and patient interviews conducted during follow-up visits. Events of interest included post-injection flare pain, joint swelling, local inflammation, or systemic symptoms. All adverse events were reviewed by the clinical team, and their relationship to the injection was determined by the treating physician.

### 2.5. Statistical Analysis

Descriptive statistics are presented as mean ± standard deviation for continuous measures and as frequency (proportion) for categorical measures. Normality of continuous variables was assessed using the Shapiro–Wilk test. Between-group comparisons for normally distributed continuous variables were performed using independent *t*-tests, and the Wilcoxon rank-sum test was used when normality was not satisfied. Categorical variables were compared using the chi-square or Fisher’s exact test, depending on sample size and distribution. A two-sided *p* < 0.05 was considered statistically significant for secondary and exploratory analyses. All statistical analyses were performed using SAS (version 9.4; SAS Institute Inc., Cary, NC, USA).

Non-inferiority analysis was conducted using the mean change from baseline in VAS and WOMAC scores. The between-group difference (HA minus collagen) and its 95% confidence interval (CI) were calculated using PROC TTEST in SAS. Non-inferiority (one-sided) was concluded if the lower bound of the 95% CI was above the negative value of the predefined margin. Sample size was determined based on a non-inferiority design to compare the efficacy of intra-articular collagen injections with that of HA in alleviating KOA pain. The null hypothesis assumed the superiority of HA over collagen. The non-inferiority margin was predefined as 1.99 on the VAS, based on prior clinical relevance [24]. The sample size calculation was based on the following parameters: an assumed standard deviation (SD) of 2.34, a power (1–β) of 80%, and a two-sided significance level (α) of 0.05 [25]. Accordingly, the required sample size was estimated at 18 patients per group to adequately demonstrate non-inferiority, defined as the upper bound of the 95% CI for the mean difference remaining below the non-inferiority margin. To enhance the robustness of the findings and to utilize the full extent of the available dataset, the final sample size was set at 20 patients per group. For WOMAC scores, non-inferiority margins were established using previously validated thresholds for minimal clinically important difference (MCID) and substantial clinical benefit (SCB), ensuring that any observed differences would be clinically relevant rather than statistically insignificant. Specifically, the margins applied for the MCID and SCB were as follows: for total WOMAC, 16.1 and 25.3; for the WOMAC pain subscale, 4.2 and 6.4; for stiffness, 1.9 and 2.6; and for function, 10.1 and 16.4, respectively [26].

## 3. Results

### 3.1. Baseline Characteristics

Baseline characteristics such as age, sex, BMI, and KL grade were statistically analyzed and confirmed to have no significant between-group differences, thereby supporting baseline group equivalence. The mean age was 67.3 ± 10.3 and 69.0 ± 9.0 years in the collagen and HA groups, respectively (*p* = 0.737; Table 1).

### 3.2. Primary Outcome: Non-Inferiority of VAS and WOMAC Score Changes

The primary outcome involved changes in VAS and WOMAC scores from baseline using non-inferiority testing. Across all VAS domains—resting, walking, and night pain—the collagen group demonstrated non-inferiority to the HA group at both the first and second injection timepoints. All comparisons yielded statistically significant results (*p* < 0.05), and the lower bounds of the 95% CIs were above the predefined non-inferiority margin, confirming clinical non-inferiority (Table 2).

Changes in WOMAC total, pain, stiffness, and function scores showed significant improvement in the collagen group, and all 95% CIs remained well within the established non-inferiority margins. These results were consistent across all domains (all *p* < 0.0001) (Table 3).

### 3.3. Secondary Outcome: VAS and WOMAC Scores

At 6 months after the first and second injections, VAS scores for resting, walking, and night pain significantly improved from the baseline in each treatment group (all *p* < 0.05). There were no significant between-group differences in VAS scores of the HA and collagen groups at any corresponding time point (*p* > 0.05) (Figure 1). Similarly, all WOMAC subscores—including pain, stiffness, and physical function—demonstrated significant within-group improvements from baseline at both post-injection assessments (6 months after the first and second injection cycles). Intergroup comparisons revealed no significant differences in WOMAC subscores at any follow-up time point (*p* > 0.05 for all) (Figure 2).

### 3.4. Patient Satisfaction

Patient-reported satisfaction was evaluated at 6 months following both the first and second injections. After the first injection, all patients reported some degree of improvement. In the collagen group, 45% and 55% of patients reported slight and significant improvement, respectively; in the HA group, 50% each reported slight and significant improvement, albeit without significant between-group distributions (*p* = 0.752). At 6 months after the second injection, 85–90% of patients in both groups reported sustained improvement, with comparable satisfaction levels and no significant differences (*p* = 0.889) (Table 4).

### 3.5. Adverse Events

Adverse events were infrequent and mild in both groups. In the collagen group, four patients (20%) experienced transient symptoms, such as injection-site pain flare or swelling. In the HA group, five patients (25%) reported similar reactions. All adverse effects resolved with conservative management; no serious complications were recorded. No significant between-group difference in adverse event rates was noted (*p* = 0.705).

## 4. Discussion

This study involved a novel systematic evaluation of the efficacy and safety of repeated collagen injections over a 1-year follow-up period and compared their performance with that of HA. We showed that intra-articular porcine-derived atelocollagen injection generated clinical outcomes that were non-inferior to those of HA over two treatment cycles in symptomatic KOA. Both treatment groups demonstrated significant improvements in pain and joint function (in VAS and WOMAC scores) without significant between-group differences in therapeutic effects. Importantly, the adverse event rate was comparable for the collagen and HA groups, which suggests that collagen, administered in two 6-monthly cycles, is a clinically safe, effective alternative to HA.

HA was selected as the comparator based on clinical relevance and methodological rigor [11], as it is the most extensively used and studied non-operative intervention for KOA and is frequently recommended as a first-line intra-articular therapy in international guidelines [27]. Despite ongoing debates regarding its long-term efficacy, HA remains the benchmark against which novel intra-articular agents are evaluated [11]. We employed a non-inferiority, rather than superiority, framework in our statistical analysis to determine whether collagen could achieve comparable outcomes to HA within an acceptable clinical margin, which reflects our pragmatic clinical question: given that HA is considered safe and modestly effective, a new treatment with similar benefits and an acceptable safety profile—while possibly conferring additional advantages, including improved structural integration or prolonged joint surface protection—could be clinically valuable [28].

Across both treatment arms, VAS scores for rest, walking, and nocturnal pain declined significantly from baseline to the end of the second cycle, reflecting a robust reduction in subjective pain burden. All WOMAC subdomains—pain, stiffness, and physical function—improved over time in both groups, and total WOMAC scores nearly halved at follow-up completion. Thus, both collagen and HA effectively address multiple dimensions of symptomatic KOA, including nociceptive pain, joint mobility, and function [11,14]. Our findings are consistent with previous reports of beneficial effects of both HA and collagen in KOA [11,14]. Multiple randomized controlled trials and meta-analyses have shown that HA provides modest pain relief and functional improvement, despite variable magnitude and effect duration [11]. Collagen-based intra-articular injections, although less studied, have gained attention in recent years [14]. In a multicenter retrospective study, Volpi et al. [29] confirmed the safety and tolerability of three intra-articular injections of low-molecular-weight (<3 kDa) bovine-derived hydrolyzed collagen. Their findings, based on a large patient cohort, provided stronger evidence than the earlier study by De Luca et al. [30], without significant adverse effects. In a double-blind, randomized, active-controlled trial by Martin et al. [20], intra-articular type I hydrolyzed porcine collagen was compared to HA in patients with KOA. At both 3- and 6-month follow-ups, collagen injections demonstrated comparable efficacy and safety to HA, suggesting that collagen may serve as a viable alternative treatment for KOA [20]. A multicenter, randomized, double-blind study by Lee et al. [31] to evaluate intra-articular type I atelocollagen in KOA and other cartilage defects showed no significant difference in VAS scores between the collagen and placebo groups at 4 and 12 weeks, but it found a significant improvement in the collagen group at 24 weeks. This is one of the first studies to directly compare two full cycles of collagen and HA injections in a real-world setting to generate novel data regarding long-term use and repeat-dose tolerability.

More than 90% of our participants in both groups reported either “slight improvement” or “marked improvement” following each injection cycle. Although a small number of patients reported no change after the second injection, none indicated worsening of symptoms, and satisfaction ratings remained stable across cycles. The similarity in satisfaction trends between groups underscores that collagen therapy was perceived as at least as beneficial as HA by the patients themselves, which further supports its clinical acceptability. Lee et al. [31] demonstrated higher patient and physician satisfaction rates in the collagen group, suggesting that intra-articular atelocollagen may effectively alleviate KOA pain and related cartilage disorders. HA products have a long-established safety profile and are routinely administered at multiple intervals without serious cumulative toxicity [11]. To address concerns regarding repeat-dose safety, we specifically designed our study to include two consecutive cycles of treatment and compared collagen against HA in this context. Our findings revealed no significant intergroup difference in the frequency or severity of adverse events, with only minor, self-limiting complaints such as transient joint discomfort and swelling. These results support the tolerability of collagen over a repeated dosing schedule and position it as a feasible long-term injectable strategy for KOA [14].

The mechanisms underlying the observed clinical improvements differ between HA and collagen, which may have implications for personalized treatment selection. HA acts primarily by restoring the viscoelastic properties of synovial fluid, thereby improving joint lubrication and absorbing mechanical shocks during movement [32,33]. HA exhibits anti-inflammatory and analgesic properties through modulation of nociceptive signaling, reduction in cytokine activity (e.g., interleukin-1β, tumor necrosis factor-α), and inhibition of metalloproteinase-mediated cartilage degradation [32,33]. However, the intra-articular half-life of HA is relatively short, and repeated administrations are frequently needed to maintain its clinical effects [11]. Collagen, a key structural protein comprising up to 75% of the dry weight of connective tissues like cartilage, plays a crucial role in supporting tissue architecture and cellular function [16]. In vitro studies showed that collagen preparations can enhance chondrocyte proliferation, stimulate extracellular matrix and HA production, and suppress inflammatory mediators [34,35]. Animal studies, including those using purified porcine atelocollagen and collagen tripeptides, demonstrated reduced inflammation, promotion of tissue repair, and delayed cartilage degeneration, suggesting a potential therapeutic role of collagen in cartilage regeneration and OA management [16,36]. Although HA primarily addresses the mechanical symptoms of OA, collagen may exert a dual function—providing symptomatic relief and potentially mitigating cartilage degradation. This hypothesis aligns with our observation that collagen-treated patients exhibited sustained improvements after two injection cycles with a low incidence of side effects.

Moreover, from a healthcare systems and policy standpoint, availability and cost-effectiveness are crucial when introducing a new biologic agent into clinical practice. Currently, collagen injections are less widely available and are associated with higher costs than standard HA formulations in many healthcare settings [14]. However, if ongoing and future studies continue to support the chondroprotective and potentially structure-modifying effects of atelocollagen, its long-term utility may justify the cost, especially in patients with moderate-stage OA aiming to postpone surgical intervention [14,31]. Furthermore, in cases of HA intolerance or suboptimal response, collagen may serve as a clinically meaningful and biologically distinct alternative [12,13,14,31]. Thus, collagen has the potential to shift the therapeutic paradigm, provided its biological advantages are substantiated in larger-scale, prospective studies [12,13,14,31].

Despite its strengths, including rigorous outcome assessment, standardized dosing intervals, and direct comparison with a gold-standard therapy, our study has some limitations. First, the majority of patients enrolled were women, which may limit the generalizability of the findings to male populations. However, this reflects the epidemiological characteristics of KOA in Asian populations, where the disease shows a markedly higher prevalence among women [37,38,39,40,41]. Second, the retrospective study design incurs an inherent risk of selection bias and residual confounding. Despite comparable between-group baseline characteristics, unmeasured factors such as activity level, adherence to rehabilitation, or prior injection response could have influenced outcomes. Third, the sample size was relatively small (*n* = 40), limiting the statistical power to detect subtle between-group differences. However, the sample size was prospectively calculated using a validated non-inferiority model, and our final sample of 20 patients per group met and exceeded the minimum requirement for robust statistical comparison. Larger studies are needed to validate our findings and explore subgroup effects based on disease severity, comorbidities, or imaging characteristics. Fourth, while we assessed efficacy over two injection cycles (approximately 12 months), the long-term durability of these benefits beyond 1 year remains unknown. Structural outcomes such as radiographic progression, cartilage volume, or biomarkers of cartilage turnover were not evaluated. Fifth, the study was not blinded, which may have introduced expectation bias, particularly in the subjective satisfaction assessments. However, this reflects real-world conditions, where blinding is frequently unfeasible. Sixth, the cohort consisted exclusively of Korean individuals, which may constrain the applicability of our findings to other racial or ethnic groups with different cartilage biology or injection response profiles. Future studies should include more diverse populations to improve generalizability. Lastly, while we employed validated tools such as VAS and WOMAC [22,23], we did not include imaging or biochemical markers to assess potential disease-modifying effects of collagen, which could be an important direction for future research.

## 5. Conclusions

This study provides preliminary evidence that intra-articular collagen injection is a safe, effective treatment for KOA that ensures pain relief and functional gains comparable to those of HA over a two-cycle regimen. Collagen represents a viable alternative injectable therapy for non-surgical intervention in KOA, especially for those with refractory KOA or intolerance of HA.

## Figures and Tables

**Figure 1 bioengineering-12-00710-f001:**
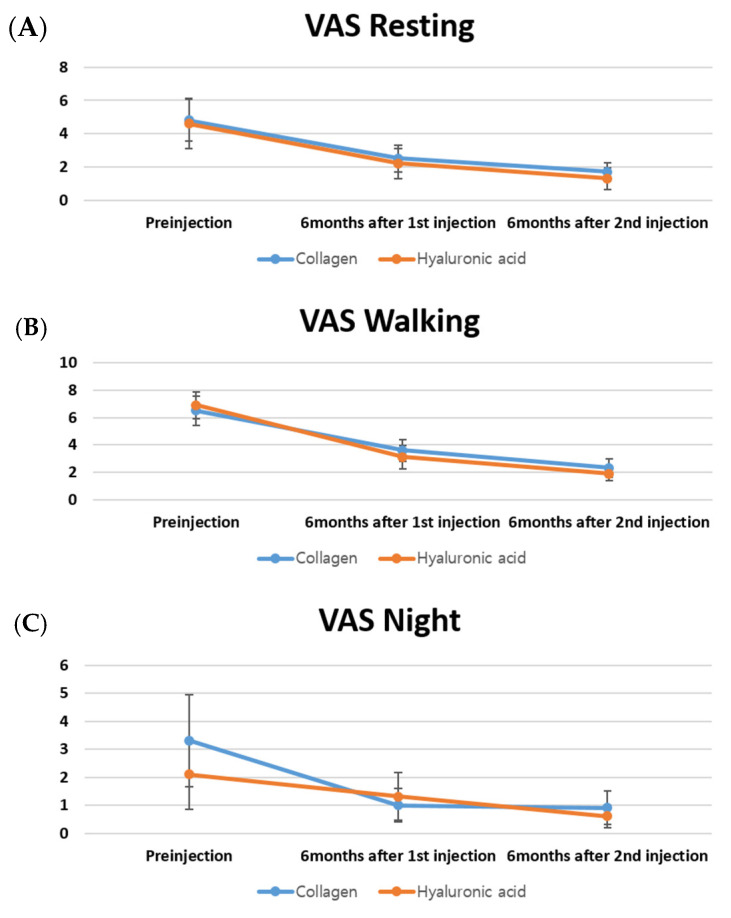
Changes in the visual analog scale (VAS) scores for resting pain (**A**), walking pain (**B**), and night pain (**C**) at baseline, 6 months after the first injection, and 6 months after the second injection in the hyaluronic acid (HA) and collagen groups.

**Figure 2 bioengineering-12-00710-f002:**
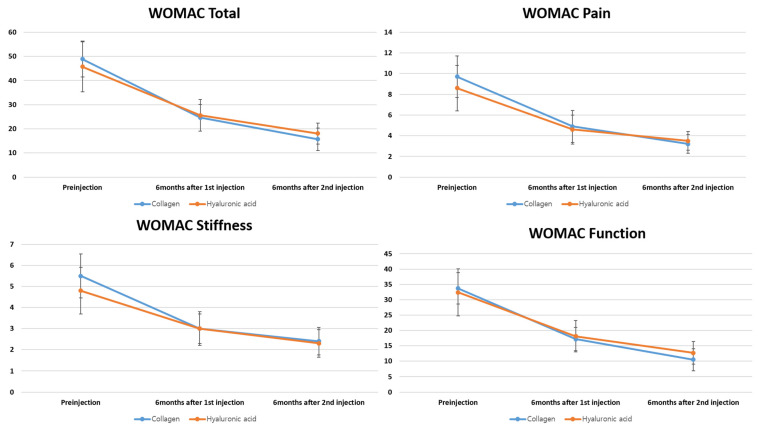
Changes in the WOMAC subscores (pain, stiffness, and physical function) at baseline, 6 months after the first injection, and 6 months after the second injection in the HA and collagen group.

**Table 1 bioengineering-12-00710-t001:** Baseline clinicodemographic characteristics of the study participants.

	Collagen	Hyaluronic Acid	*p*-Value
Age, years	67.3 ± 10.3	69.0 ± 9.0	0.737
Sex, female/male, n	15/5	16/4	1
BMI, kg/m^2^	23.5 ± 3.8	24.8 ± 3.2	0.257
Occupation (n, %)			0.305
Housewife	10 (50%)	14 (70%)	
White collar job	4 (20%)	1 (5%)	
Pink-collar job	2 (10%)	0 (0%)	
Blue-collar job	0 (0%)	3 (15%)	
Inoccupation	4 (20%)	2 (10%)	
Kellgren–Lawrence OA grade	2.8 ± 1.0	2.35 ± 0.99	0.151
II (n, %)	7 (35%)	10 (50%)	0.337
III (n, %)	13 (65%)	10 (50%)	
Previous medication history (n, %)			0.500
NSAIDs	12 (60%)	10 (50%)	
NSAIDs + SYSADOAs	6 (30%)	9 (45%)	
NSAIDs + SYSADOAs + Analgesics	2 (10%)	1 (5%)	
Previous injection history (n, %)			0.100
Yes	11 (55%)	6 (30%)	
No	9 (45%)	14 (70%)	
Injection site (Rt, %)	14 (70%)	18 (90%)	0.235
Underlying disease (n, %)			
Hypertension	11 (55%)	10 (50%)	0.751
Diabetes	4 (20%)	5 (25%)	1
Thyroid disease	4 (20%)	0 (0%)	0.106
Smoking (n, %)	6 (30%)	6 (30%)	1
Pre-injection			
VAS resting	4.8 ± 2.5	4.6 ± 3.0	0.776
VAS walking	6.5 ± 2.1	6.9 ± 2.0	0.538
VAS night	3.3 ± 3.3	2.1 ± 2.5	0.299
Total WOMAC	48.9 ± 14.9	45.7 ± 20.6	0.583
Pain WOMAC	9.7 ± 4.0	8.6 ± 4.4	0.438
Stiffness WOMAC	5.5 ± 2.1	4.8 ± 2.2	0.281
Function WOMAC	33.7 ± 10.3	32.4 ± 15.3	0.745

Numerical variables are presented as the mean ± SD, and categorical variables as the frequency (proportion). BMI, body mass index; OA, osteoarthritis; NSAIDs, non-steroidal anti-inflammatory drugs; SYSADOAs, symptomatic slow-acting drugs for osteoarthritis; VAS, visual analog scale; WOMAC, the Western Ontario and McMaster Universities Osteoarthritis Index.

**Table 2 bioengineering-12-00710-t002:** Evaluation of non-inferiority using the visual analog scale scores in the collagen and hyaluronic acid groups.

	Collagen	Hyaluronic Acid	Non-Inferiority Test (Margin: VAS = 1.99)	
			*p*-Value	95% CI
Pre-injection–1st injection				
VAS resting	2.3 ± 2.0	2.4 ± 2.2	0.004	(−1.218, ∞)
VAS walking	3.0 ± 2.5	3.2 ± 2.1	0.010	(−1.433, ∞)
VAS night	1.8 ± 2.3	1.1 ± 1.4	<0.0001	(−0.312, ∞)
Pre-injection–2nd injection				
VAS resting	3.1 ± 2.1	3.3 ± 2.6	0.011	(−1.459, ∞)
VAS walking	4.4 ± 2.4	4.7 ± 2.5	0.019	(−1.635, ∞)
VAS night	2.4 ± 2.6	1.6 ± 2.0	0.0003	(−0.440, ∞)

Numerical variables are presented as mean ± SD. VAS, visual analog scale; WOMAC, the Western Ontario and McMaster Universities Osteoarthritis Index. ∞ indicates that the upper bound of the confidence interval is not limited.

**Table 3 bioengineering-12-00710-t003:** Evaluation of non-inferiority using the WOMAC scores of the collagen and hyaluronic acid groups.

	Collagen	Hyaluronic Acid	Non-Inferiority Test (Margin: WOMAC = MCID)		Non-Inferiority Test (Margin: WOMAC = SCB)	
			*p*-Value	95% CI	*p*-Value	95% CI
Pre-injection–1st injection						
Total WOMAC	24.3 ± 11.0	20.1 ± 10.4	<0.0001	(−1.502, ∞)	<0.0001	(−1.502, ∞)
Pain WOMAC	4.8 ± 2.6	4.0 ± 2.6	<0.0001	(−0.634, ∞)	<0.0001	(−0.634, ∞)
Stiffness WOMAC	2.5 ± 1.7	1.8 ± 1.3	<0.0001	(−0.043, ∞)	<0.0001	(−0.043, ∞)
Function WOMAC	16.5 ± 7.8	14.2 ± 7.3	<0.0001	(−1.724, ∞)	<0.0001	(−1.724, ∞)
Pre-injection–2nd injection						
Total WOMAC	33.2 ± 12.8	27.6 ± 14.9	<0.0001	(−1.858, ∞)	<0.0001	(−1.858, ∞)
Pain WOMAC	6.8 ± 3.1	5.4 ± 3.4	<0.0001	(−0.384, ∞)	<0.0001	(−0.384, ∞)
Stiffness WOMAC	3.1 ± 1.9	2.5 ± 1.9	<0.0001	(−0.372, ∞)	<0.0001	(−0.372, ∞)
Function WOMAC	23.2 ± 8.9	19.7 ± 10.5	<0.0001	(−1.651, ∞)	<0.0001	(−1.651, ∞)

Numerical variables are presented as mean ± SD. VAS, visual analog scale; WOMAC, the Western Ontario and McMaster Universities Osteoarthritis Index. ∞ indicates that the upper bound of the confidence interval is not limited.

**Table 4 bioengineering-12-00710-t004:** Patient satisfaction levels stratified by group and time point.

	Collagen	Hyaluronic Acid	*p*-Value
Six months after the 1st injection			0.752
No change (n, %)	0(0%)	0(0%)	
Slightly improved (n, %)	9(45%)	10(50%)	
Very improved (n, %)	11(55%)	10(50%)	
Six months after the 2nd injection			0.889
No change (n, %)	2(10%)	3(15%)	
Slightly improved (n, %)	15(75%)	14(70%)	
Very improved (n, %)	3(15%)	3(15%)	

## Data Availability

The data published in this research are available on request from the corresponding author (MSK).

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
