# Peer review of "Clinical Efficacy and Safety of Two Cycles of Intra-Articular Injection of Porcine Atelocollagen Versus Hyaluronic Acid in Knee Osteoarthritis"

_bioengineering, 2025, doi:10.3390/bioengineering12070710_

Round 1

Reviewer 1 Report

Comments and Suggestions for Authors

The article bioengineering-3710451 presents a comparative analysis of two biological agents for the treatment of KOA. The topic is relevant within the context of seeking alternatives to hyaluronic acid. The title and abstract fully correspond to the manuscript's content.

The work clearly states the objective and the hypothesis of collagen's non-inferior efficacy; moreover, validated scales (WOMAC, VAS) were used, aligning with KOA assessment standards. The statistical approach (non-inferiority analysis) is appropriate for the stated task. However, the retrospective design and lack of randomization limit the strength of the conclusions.

The methodology does not specify how the comparability of groups regarding baseline characteristics (KOA stage by Kellgren-Lawrence, BMI, age, sex) was ensured.

There is also no information on the patient allocation method (what determined the choice of collagen/ hyaluronic acid?).

I ask the authors to provide more arguments supporting the article's relevance. As the authors themselves note, similar studies have been conducted before, for example [27]. A detailed justification of the study's novelty should be included in the Introduction.

For Figures 1 and 2, I request the visualization of error bars on the graph.

Currently, bioengineering-3710451  is a well-written work, and the authors' conclusion aligns with the manuscript text, but overall, the article lacks novelty.

I suggest the authors conduct a more thorough comparison of both approaches in terms of their availability and cost-effectiveness and perform a more comprehensive analysis. If the authors can demonstrate the relevance and novelty of the study, I recommend accepting the article after revision.

Author Response

Comments 1: The article bioengineering-3710451 presents a comparative analysis of two biological agents for the treatment of KOA. The topic is relevant within the context of seeking alternatives to hyaluronic acid. The title and abstract fully correspond to the manuscript's content.

: We thank the reviewer for the constructive and insightful comments, which significantly helped improve the quality and clarity of our manuscript.

Comments 2: The work clearly states the objective and the hypothesis of collagen's non-inferior efficacy; moreover, validated scales (WOMAC, VAS) were used, aligning with KOA assessment standards. The statistical approach (non-inferiority analysis) is appropriate for the stated task.

: We thank the reviewer for the constructive and insightful comments, which significantly helped improve the quality and clarity of our manuscript.

Comments 3: However, the retrospective design and lack of randomization limit the strength of the conclusions.

: We fully agree with the reviewer that the retrospective, non-randomized design inherently limits the ability to draw causal conclusions. To address this limitation, we have now clearly acknowledged this point in the Discussion section. (Lines 354-358)

Comments 4: The methodology does not specify how the comparability of groups regarding baseline characteristics (KOA stage by Kellgren-Lawrence, BMI, age, sex) was ensured.

: Thank you for this important point. In the Methods and results section, we have added the following clarification: “To ensure comparability between the two groups, we applied the same inclusion and exclusion criteria, and selected patients consecutively over the same time period. Baseline characteristics such as age, sex, BMI, and Kellgren–Lawrence grade were statistically analyzed and confirmed to have no significant between-group differences (Table 1), thereby supporting baseline group equivalence.” (Lines 92-94, 184-186) This ensures transparency in patient selection and strengthens the internal validity of the findings.

Comments 5: There is also no information on the patient allocation method (what determined the choice of collagen/ hyaluronic acid?).

: We thank the reviewer for raising this important point. The choice between collagen and HA injection was not randomized but determined through a shared decision-making process between the patient and the orthopedic surgeon. All patients included in this study had radiographically confirmed Kellgren–Lawrence (KL) grade 2 or 3 knee osteoarthritis, which reflects mild to moderate disease severity—an appropriate stage for injectable therapy. We have now added the following clarification in the Methods section. (Lines 102-109) This clarification reflects real-world clinical practice and provides transparency regarding allocation decisions in this retrospective study.

Comments 6: I ask the authors to provide more arguments supporting the article's relevance. As the authors themselves note, similar studies have been conducted before, for example [27]. A detailed justification of the study's novelty should be included in the Introduction.

: We appreciate the reviewer highlighting the need to better articulate novelty. We have revised the Introduction section (Lines 61–66) to more clearly state the gap addressed by our study: “Although previous trial has compared collagen and HA, this study often involved only a single injection cycle or had limited follow-up. Our study is, to the best of our knowledge, the first to perform a comparative evaluation of two complete injection cycles of intra-articular collagen versus HA in a real-world clinical setting. We also conducted a formal non-inferiority analysis, which has not been rigorously applied in prior comparative studies.” This distinction highlights the contribution of our study to the existing literature.

Comments 7: For Figures 1 and 2, I request the visualization of error bars on the graph.

: Thank you for the helpful suggestion. We have updated Figures 1 and 2 to include standard deviation error bars for each data point to better illustrate variability and statistical reliability. These revised figures have been incorporated into the revised manuscript.

Comments 8: Currently, bioengineering-3710451 is a well-written work, and the authors' conclusion aligns with the manuscript text, but overall, the article lacks novelty. I suggest the authors conduct a more thorough comparison of both approaches in terms of their availability and cost-effectiveness and perform a more comprehensive analysis. If the authors can demonstrate the relevance and novelty of the study, I recommend accepting the article after revision.

: We appreciate the reviewer’s comment regarding the need to emphasize novelty and expand the discussion on availability and cost-effectiveness. First, regarding novelty, while prior studies such as Martin et al. have compared collagen and HA, our study offers several key distinctions: We conducted a two-cycle comparative analysis, while most previous trials were limited to single-cycle or short-term designs. We applied a non-inferiority statistical framework with predefined clinical margins, which is rarely applied in collagen-related studies and reflects a robust methodological design aligned with real-world therapeutic decision-making. Unlike prior RCTs that often exclude complex cases, our study reflects a real-world Korean cohort using consistent inclusion/exclusion criteria across groups, thus providing translatable insights into routine orthopedic practice. (Lines 61-66) These elements contribute to the growing but still limited body of evidence supporting repeated intra-articular collagen injections and provide clinicians with practical reference data on treatment durability and patient satisfaction across multiple timepoints.

Second, with respect to availability and cost-effectiveness, we agree this is a critical consideration for clinical translation. We have now expanded our Discussion section (Lines 338–348) to include: “From a healthcare systems perspective, collagen injections are currently less widely available and more costly than HA in many regions. However, collagen may offer greater biological potential for cartilage support and chondroprotection, especially for patients who are HA-intolerant or unresponsive. As such, if future studies validate long-term structural benefits, the cost-effectiveness profile of collagen could be favorably shifted—particularly in younger patients or those seeking to delay joint replacement surgery.” This addition situates collagen not only as a symptomatic treatment, but also as a strategically viable option in the broader OA management spectrum, contingent on patient-specific factors. We hope this more comprehensive discussion better addresses the reviewer’s request and underscores the clinical relevance and forward-looking potential of our findings.

We have revised the manuscript to reflect all the reviewer’s suggestions, including clarifying methodology, enhancing the rationale and novelty, updating figures, and expanding the discussion on real-world relevance and cost. We believe these revisions substantially improve the manuscript and hope it now meets the reviewer’s expectations. Once again, we thank the reviewer for their careful evaluation and valuable input.

Reviewer 2 Report

Comments and Suggestions for Authors

In this manuscript, the authors describe the "Clinical Efficacy and Safety of Two Cycles of Intra-articular Injection of Porcine Atelocollagen versus Hyaluronic Acid in Knee Osteoarthritis". It would be helpful if the authors address the concerns below'

1) While this comparative study of the efficacy of hyaluronic acid versus two cycles of Intra-articular Injection of Porcine Atelocollagen over a specified period is quite plausible, the pool of subjects in this study is quite narrow (as acknowledge in line 326) and thereby making deductions from this study less valid. It would be helpful to readers if the authors increase the pool of subjects in this study. 

2) This study would also be beneficial to readers if it is expanded to include subjects from different races.

3) While this study laid out a logical point based on data regarding the effectiveness of collagen injections as a viable alternative to hyaluronic acid administration, the authors did not disclose the fitness status of the subjects involved, the kind of medications they were on which could potentially influence or alter the outcome of this study. It would be helpful if the authors provide an insight regarding this.

4) While Chondroitin Sulfate and glucosamine has been known to be generally effective in managing issues of osteoarthritis, why was this study not compared with Chondroitin sulfate/glucosamine in Knee Osteoarthritis? It will be helpful to readers if the authors address this.

In summary, this manuscript could potentially be beneficial to its target readers if the above concerns are adequately addressed.

Author Response

Comments 1: In this manuscript, the authors describe the "Clinical Efficacy and Safety of Two Cycles of Intra-articular Injection of Porcine Atelocollagen versus Hyaluronic Acid in Knee Osteoarthritis". It would be helpful if the authors address the concerns below'

While this comparative study of the efficacy of hyaluronic acid versus two cycles of Intra-articular Injection of Porcine Atelocollagen over a specified period is quite plausible, the pool of subjects in this study is quite narrow (as acknowledge in line 326) and thereby making deductions from this study less valid. It would be helpful to readers if the authors increase the pool of subjects in this study. 

: Thank you for this important observation. We fully acknowledge that the sample size of our study is relatively limited, which may restrict the generalizability of the findings. Accordingly, we have revised the Limitations section to explicitly reflect this point. (Lines 358-363) Furthermore, we would like to emphasize that our study employed a formal sample size calculation based on a non-inferiority design, which determined that a minimum of 18 participants per group would be required to achieve 80% power and a two-sided α of 0.05. To enhance robustness, we ultimately included 20 patients in each group, which satisfies the statistical requirements for our primary outcome. (Lines 166-174) This methodological rigor supports the internal validity of our findings, although we concur that larger, multicenter studies are needed for broader external validation.

Comments 2:  This study would also be beneficial to readers if it is expanded to include subjects from different races.

: We appreciate the reviewer’s thoughtful suggestion. Indeed, our cohort consisted solely of Korean patients, reflecting the demographic characteristics of our institution's catchment area. We recognize that this homogeneity may limit extrapolation of the results to multiethnic populations. As such, we have amended the Limitations section to include the following: "The cohort consisted exclusively of Korean individuals, which may constrain the applicability of our findings to other racial or ethnic groups with different cartilage biology or injection response profiles. Future studies should include more diverse populations to improve generalizability." (Lines 369-372)

Comments 3: While this study laid out a logical point based on data regarding the effectiveness of collagen injections as a viable alternative to hyaluronic acid administration, the authors did not disclose the fitness status of the subjects involved, the kind of medications they were on which could potentially influence or alter the outcome of this study. It would be helpful if the authors provide an insight regarding this.

: We thank the reviewer for this important and constructive comment. While our study did not directly measure objective physical fitness scores, all enrolled participants were ambulatory and assessed by the treating physician to be clinically fit for outpatient intra-articular injection therapy. To provide a surrogate marker for baseline physical function, we have additionally included occupational activity levels (i.e., occupational profiles) in our dataset to reflect functional status and physical capacity in daily life.

Moreover, we have expanded our data collection to include detailed medication histories and prior joint interventions. Specifically, we reviewed the use of oral analgesics, non-steroidal anti-inflammatory drugs (NSAIDs), symptomatic slow-acting drugs for osteoarthritis (SYSADOAs; including chondroitin sulfate and glucosamine), and documented histories of previous intra-articular injections (including corticosteroids and HA). These variables were compared between groups, and no statistically significant differences were observed, indicating comparable baseline exposure to medications and interventions. These additions have been reflected in both the Methods and Results sections of the revised manuscript. (Lines 130-133 and Table 1) We believe that these modifications improve the transparency and methodological robustness of the study and adequately address the reviewer’s concern.

Comments 4: While Chondroitin Sulfate and glucosamine has been known to be generally effective in managing issues of osteoarthritis, why was this study not compared with Chondroitin sulfate/glucosamine in Knee Osteoarthritis? It will be helpful to readers if the authors address this.

: We sincerely thank the reviewer for this insightful comment. Our study specifically aimed to compare two intra-articular injection therapies—porcine-derived atelocollagen and hyaluronic acid (HA)—based on their local delivery mechanism, similar administration protocols, and shared objective of directly modulating the joint environment in patients with knee osteoarthritis (KOA). Chondroitin sulfate and glucosamine, as representative symptomatic slow-acting drugs for osteoarthritis (SYSADOAs), are commonly used oral agents with systemic action and distinct pharmacodynamic profiles, and thus were not included as comparators in this intra-articular-focused study.

However, we acknowledge the importance of accounting for the potential influence of SYSADOA use on clinical outcomes. Accordingly, we investigated the prior and concurrent use of SYSADOAs—including chondroitin sulfate and glucosamine—alongside other medications such as NSAIDs and oral analgesics. This information has been added to the revised Methods and Results sections. Importantly, there were no statistically significant differences in SYSADOA use between the collagen and HA groups, indicating minimal confounding from this variable. This methodological step further supports the internal validity of our treatment effect comparisons. (Lines 130-133 and Table 1)

Comments 5: In summary, this manuscript could potentially be beneficial to its target readers if the above concerns are adequately addressed.

: We greatly appreciate the reviewer’s thoughtful overall assessment and constructive suggestions. We have carefully reviewed all comments and have revised the manuscript accordingly to address each point in detail. We sincerely hope that these revisions strengthen the scientific rigor, clarity, and clinical relevance of the manuscript, and make it more valuable to the readers of the journal.

Reviewer 3 Report

Comments and Suggestions for Authors

The present clinical study compares in a rigorous design the safety and effycacy of hyaluronic acid and collagen injected into the joint of patients in two cycles. Although the effects were shown to be good in both cases, surprisingly the study did not show differences in effect after administration of the different agents. In any case, this is an important and significant work confirming the quality of the current treatment, which is being considered as a standard for comparing the effects of newly developed formulations. An important discussion is built on the results obtained and appropriately supplemented with hypotheses as to why such results were likely to have been obtained and what further research might be needed. I have no comments on the very well written manuscript and agree with the publication as it is.

Author Response

Comments 1: The present clinical study compares in a rigorous design the safety and effycacy of hyaluronic acid and collagen injected into the joint of patients in two cycles. Although the effects were shown to be good in both cases, surprisingly the study did not show differences in effect after administration of the different agents. In any case, this is an important and significant work confirming the quality of the current treatment, which is being considered as a standard for comparing the effects of newly developed formulations. An important discussion is built on the results obtained and appropriately supplemented with hypotheses as to why such results were likely to have been obtained and what further research might be needed. I have no comments on the very well written manuscript and agree with the publication as it is.

: We sincerely thank the reviewer for the thoughtful and encouraging comments on our manuscript. We appreciate the recognition of our study's design and the clinical relevance of comparing intra-articular collagen and hyaluronic acid injections across two treatment cycles. As the reviewer noted, our findings affirm the non-inferiority of collagen to HA in symptomatic management of KOA and support its potential as a viable alternative in routine orthopedic care. We are especially grateful for your acknowledgment of the importance of our discussion, which aimed to provide not only interpretation but also meaningful clinical context and hypotheses for future exploration. Although no significant differences were observed between the treatment arms, we agree with the reviewer that these findings reinforce the validity of HA as a clinical benchmark and highlight the need for further long-term studies on potential structure-modifying effects and cost-effectiveness of collagen therapy. Thank you once again for your positive feedback and support of the publication.